# HT-Transformer: Event Sequences Classification by Accumulating Prefix Information with History Tokens

## Abstract

Deep learning has achieved strong results in modeling sequential data, including event sequences, temporal point processes, and irregular time series. Recently, transformers have largely replaced recurrent networks in these tasks. However, transformers often underperform RNNs in sequence classification tasks that aim to predict future targets. The reason behind this performance gap remains largely unexplored. In this paper, we identify a key limitation of transformers: the absence of a single state vector that provides a compact and effective representation of the entire sequence. Additionally, we show that contrastive pretraining of embedding vectors fails to capture local context, which is crucial for accurate prediction. To address these challenges, we introduce history tokens, a novel concept that facilitates accumulating historical information during next-token prediction pretraining. Our approach significantly improves transformer-based models, achieving impressive results in finance, e-commerce, and healthcare tasks. The code is publicly available on GitHub: `https://github.com/anonymous-10647849/ht-transformer-submission`.

## 1 Introduction

Many real-world problems involve predicting future events from historical observations. In generative tasks, the goal is to forecast events similar to those previously observed (Xue et al., 2024). However, many practical applications require anticipating events not explicitly appearing in the training history. Examples include loan default, customer churn, and disease onset. These scenarios are typically addressed using classical machine learning models, such as logistic regression or gradient boosting, applied to handcrafted features or unsupervised model-based embeddings derived from historical data (Osin et al., 2024; Synerise, 2025).

Deep learning has shown significant success in modeling sequential structures, including event sequences, temporal point processes, and time series data. A prominent trend is the adoption of pretrained Transformer architectures due to their ability to capture long-range dependencies and complex temporal patterns (Padhi et al., 2021; Zuo et al., 2020). Unlike recurrent neural networks, however, Transformers lack a canonical mechanism for extracting a fixed-size embedding from a sequence, as information is distributed across the activations of all tokens. This issue is commonly mitigated through auxiliary objectives during pretraining, such as contrastive learning (BehnamGhader et al., 2024), sentence order prediction (Lan et al., 2020), or next-sentence prediction (Devlin et al., 2019). However, each of these approaches introduces limitations. For instance, it is well documented that contrastive pretraining may overemphasize "easy features", compromising downstream quality (Robinson et al., 2021). Consequently, the problem of learning robust and informative sequence embeddings, including methods based on the next-token prediction (NTP) objective (Yenduri et al., 2024), remains an open research question.

In this work, we propose a novel approach to pretraining Transformer-based embeddings without relying on auxiliary tasks. Our method draws inspiration from recurrent architectures and leverages sparse attention masks to guide the accumulation of historical information (Bulatov et al., 2022). Specifically, we introduce *history tokens* that gather and summarize contextual information during training via a standard next-token prediction objective, as illustrated in Figure 1. We empirically

(a) History Token serves as a bottleneck during pre-training.

(b) The embedding of History Token is used in downstream tasks.

Figure 1: History tokens accumulate prefix information during pretraining via next-token-prediction. The embedding of the History Token is later used in downstream tasks.

evaluate the resulting embeddings across multiple domains, including finance, e-commerce, and healthcare, and show that they offer strong predictive performance, especially for tasks focused on forecasting future events rather than global sequence classification.

The contributions of this paper are as follows:

1. We propose a novel HT-Transformer architecture that employs history tokens to accumulate past information during pretraining using only the next-token prediction objective.

2. We develop advanced strategies for history tokens position selection and attention masking for improved downstream quality.

3. We introduce strong baselines for embedding extraction by utilizing adapted variants of Recurrent Memory Transformer and Longformer.

4. We demonstrate that the proposed HT-Transformer is particularly effective for tasks focused on predicting future events and establish new state-of-the-art results across multiple benchmarks in finance, e-commerce, and healthcare.

## 2 PRELIMINARIES ON EVENT SEQUENCES

This work focuses on modeling sequences of discrete events $S = \{s_i\}_{i=1}^N$, where each event $s_i$ is represented by a collection of fields, including a timestamp $t_i$, optional numerical attributes, and categorical variables. Each sequence typically corresponds to a single entity, such as a user or client, and the events are ordered chronologically by their timestamps: $t_1 \leq t_2 \leq \cdots \leq t_N$. An illustration of such sequences is provided in Figure 2.

**Data preprocessing.** Before inputting data into a deep model, each event in the sequence must be transformed into an embedding in a latent space. In a typical preprocessing pipeline, each data field is encoded independently, and the resulting embeddings are concatenated to form a single event representation (Gorishniy et al., 2021). Categorical features are transformed by assigning a trainable embedding vector to each possible value. Numerical features are incorporated directly into the event embedding without additional preprocessing.

We apply time-based positional encoding when using Transformer models, following the approach proposed in prior work (Yang et al., 2022). Specifically, for each timestamp $t$, we compute a positional embedding $\text{PE}_i(t)$ of dimension $d$ as:

$$\text{PE}_i(t) = \begin{cases} \sin\left(t/\left(m * \left(\frac{5M}{m}\right)^{\frac{i}{d}}\right)\right), & \text{if } i \text{ is even} \\ \cos\left(t/\left(m * \left(\frac{5M}{m}\right)^{\frac{i-1}{d}}\right)\right), & \text{otherwise} \end{cases} \tag{1}$$

where $m$ and $M$ are constants determined from the distribution of timestamp values. We refer the reader to the original work for implementation details and parameter selection.

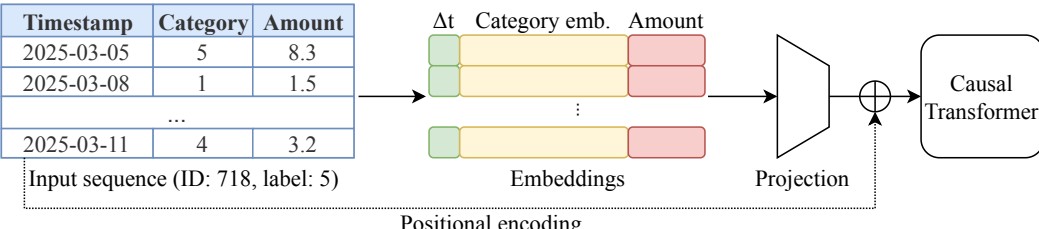

Figure 2: Event sequences preprocessing.

**Pretraining.** We consider sequence-level classification tasks, where each sequence $S$ is associated with a single target label $l$. Because labels are assigned at the sequence level rather than the event level, the number of labeled examples is often much smaller than the total number of events. This imbalance motivates the development of unsupervised pretraining algorithms that can leverage the abundance of unlabeled sequential data to improve downstream performance.

Unsupervised pretraining on event sequences is typically based on either generative or contrastive learning objectives.

In the *generative* approach, the model is trained to predict the next event $s_{i+1}$ given the historical context $s_1, \ldots, s_i$, encouraging the model to capture temporal dependencies and sequence structure. A typical generative loss is formulated as a weighted sum of individual losses over each data field (Shchur et al., 2019; Padhi et al., 2021; McDermott et al., 2023). Timestamps can be predicted using standard regression losses such as Mean Absolute Error (MAE) or Mean Squared Error (MSE), or through more expressive temporal point process models based on intensity functions (Rizoiu et al., 2017; Zuo et al., 2020). In our work, we adopt the MAE loss for timestamp prediction:

$$\mathcal{L}_{\mathrm{MAE}}(\Delta\hat{t}, \Delta t) = |\Delta\hat{t} - \Delta t|, \tag{2}$$

where $\Delta\hat{t}$ is the predicted inter-event time and $\Delta t$ is the ground truth. We apply the same MAE objective to other numerical fields and use the cross-entropy loss for categorical attributes.

An alternative to generative modeling is *contrastive learning* (Babaev et al., 2022), which aims to learn sequence representations by maximizing agreement between different augmented views of the same sequence and pushing apart views from different sequences. Typically, each sequence is divided into multiple, possibly overlapping, subsequences $R_k \subset S$ for $k = 1, \ldots, K$. Let $\mathrm{ID}(R)$ denote the index of the original sequence from which a chunk $R$ was derived. Then the contrastive loss (Chopra et al., 2005) is defined as:

$$\mathcal{L}_{\mathrm{cont}}(R_i, R_j) = \begin{cases} \|f(R_i) - f(R_j)\|^2, & \text{if } \mathrm{ID}(R_i) = \mathrm{ID}(R_j) \\ \max\left(0, \epsilon - \|f(R_i) - f(R_j)\|\right)^2, & \text{otherwise} \end{cases} \tag{3}$$

where $f(R) \in \mathbb{R}^d$ is the embedding of a subsequence $R$ produced by the model. Following prior work (Babaev et al., 2022), we use $\epsilon = 0.5$ and $K = 5$ subsequences per sequence.

Both generative and contrastive paradigms have been successfully adapted to neural architectures such as recurrent neural networks (RNNs) and Transformers. However, while effective for specific tasks, these approaches exhibit notable limitations, especially when the goal is to anticipate future events rather than to summarize past behavior. Overcoming these limitations is a key motivation behind the approach proposed in this work.

**Downstream tasks.** Embeddings obtained from a pretrained model are typically applied to downstream classification and regression tasks, either through a fully connected head (Synerise, 2025) or via gradient boosting methods (Babaev et al., 2022). Following prior work on event sequences (Osin et al., 2024), we employ gradient boosting models for classification.

## 3 PROPOSED METHOD

The core idea of the proposed method is the introduction of special *history tokens* into Transformer models. These tokens are designed to accumulate information from preceding tokens in the se-

quence. A carefully constructed attention mask ensures that these tokens act as an information bottleneck, similar in function to the hidden states in RNNs. In the following, we describe the training procedure and the application of history tokens to downstream classification tasks.

## 3.1 Unsupervised Pretraining with History Tokens

The proposed approach is compatible with any Transformer architecture that employs a causal attention mask (Fig. 3a), where each token attends only to preceding tokens. Transformers for event sequences typically have three primary components: an event embedder, a backbone, and a prediction head. History tokens are inserted into the backbone input after event embeddings are computed, as shown in Figure 1a. Each history token is assigned the timestamp of a preceding event. These timestamps are then used for positional encoding.

To enable history tokens to serve as memory units, we modify the attention mask used by the backbone. Each history token is allowed to attend to all preceding event tokens (except other history tokens), thereby accumulating prefix information. In contrast, event tokens can attend only to history tokens and to events between the current position and the most recent history token. This attention pattern is illustrated in Figure 3d. When multiple history tokens are present, we introduce two attention strategies. In the *Last* strategy, each event token is restricted to attend only to the most recent preceding history token. In the *Random* strategy, illustrated in Figure 3e, each event token randomly selects one of the prior history tokens during attention computation. As demonstrated in our experiments, the Random strategy consistently yields better performance across various tasks.

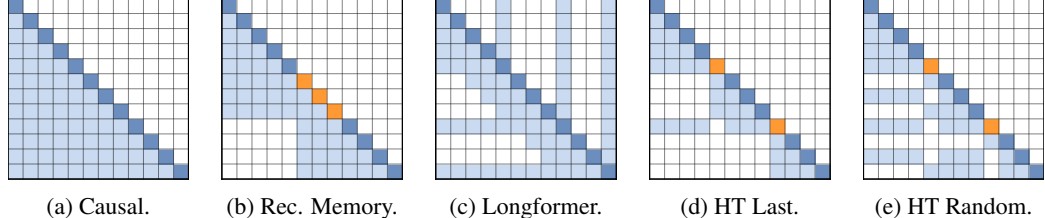

|               |                    |                   |              |                  |
| :-----------: | :----------------: | :---------------: | :----------: | :--------------: |
| (a) Causal.   | (b) Rec. Memory.   | (c) Longformer.   | (d) HT Last. | (e) HT Random.   |

Figure 3: Comparison of attention masks. Special tokens are orange-colored.

The proposed method provides considerable flexibility in selecting both the number and the placement of history tokens. For a given sequence of length $L$, the number of history tokens is computed as $\max(1, \lfloor fL \rfloor)$, where $\lfloor \cdot \rfloor$ is a rounding operator and $f$ is a tunable hyperparameter referred to as the *frequency*.

We also implement two strategies for inserting history tokens into the sequence. The *Uniform* strategy inserts history tokens at uniformly sampled positions. However, this approach can lead to a discrepancy between training and inference, as history tokens are typically positioned near the end of the sequence during evaluation. To address this issue, we introduce the *Bias-End* strategy, which places history tokens closer to the sequence's end. Specifically, it samples positions uniformly within the range $[\mu/2, L]$, where $\mu$ is the mean sequence length in the batch, and $L$ is the maximum sequence length. Our experiments show that the Bias-End strategy consistently leads to improved downstream performance.

At inference time, the history token is inserted only at the end of the sequence. In this setting, event tokens cannot access any preceding history tokens, creating a mismatch with the pretraining setup, where history tokens may appear throughout the sequence. To mitigate this discrepancy, we apply history tokens in only a subset of pretraining batches with some *application probability* $p$ (typically 50%). This partial application encourages the model to remain robust across both configurations.

## 3.2 Classification

During embedding extraction, a single history token is appended to the end of the input sequence, and the average of its hidden activations across Transformer layers is used as the sequence level embedding (Figure 1b). This embedding then serves as an input feature for a downstream gradient boosting classifier.

## 4 RELATED WORK

Transformer models have a long and successful application history to sequence classification and regression tasks, particularly in natural language processing (NLP) (Vaswani et al., 2017). One of the main challenges in this setting is the limited availability of labeled data, which has driven the development of effective unsupervised pretraining strategies (Muennighoff et al., 2023). A notable early approach is BERT (Devlin et al., 2019), which introduced a masked language modeling (MLM) objective alongside a next sentence prediction (NSP) task to enable powerful sequence representations. These pretrained models proved highly effective for downstream classification tasks such as natural language understanding (NLU) (Wang et al., 2019).

A central issue in applying Transformers to NLU is extracting a compact, semantically meaningful representation of an entire sequence. In BERT, this was addressed by introducing a special classification token trained via the NSP objective. However, subsequent work such as RoBERTa (Liu et al., 2019) challenged the necessity of the NSP task, demonstrating that it could be omitted without degrading performance.

Beyond BERT-style objectives, other works have explored contrastive pretraining techniques, such as LLM2Vec (BehnamGhader et al., 2024), which aim to bring semantically similar sequences closer in embedding space. While contrastive learning can yield strong performance when carefully implemented, it suffers from notable limitations. In particular, models can rely on "easy" features, such as surface-level similarities, to distinguish positive pairs, bypassing the need for deeper semantic understanding. Furthermore, contrastive pretraining tends to bias models toward capturing global sequence properties at the expense of local or up-to-date information. This bias poses a particular challenge in event sequence modeling, where the most recent context is often essential for accurate prediction, in contrast to many natural language processing tasks emphasizing global semantics.

Alternative methods for sequence embedding extraction include averaging token activations across specific layers or using the final token's activation (Stankevičius & Lukoševičius, 2024). However, these approaches generally underperform compared to specialized embedding pretraining techniques, particularly in tasks requiring nuanced or fine-grained representations.

In contrast, RNNs naturally summarize sequences through the hidden state at the final timestep, which compactly encodes the information needed for future prediction. This property has inspired the integration of recurrent principles into Transformer architectures, particularly for modeling long sequences. For instance, *Recurrent-Memory Transformers* (RMT) recursively apply a Transformer model to chunks of the input sequence, using special tokens to pass information between successive segments (Bulatov et al., 2022). The corresponding attention mask is shown in Figure 3b. Similar ideas are adopted in architectures such as *Longformer* (Beltagy et al., 2020), where global tokens aggregate and propagate information across extended contexts (Figure 3c). More recently, recurrent-style Transformers have been combined with contrastive learning objectives to attain strong results on natural language understanding (NLU) tasks, while retaining the generative abilities of causal models (Zhang et al., 2025).

Our work extends the Recurrent Transformer paradigm (Bulatov et al., 2022) by introducing a novel mechanism for representation learning from event sequences. We propose the use of history tokens to accumulate and summarize contextual information during NTP pretraining. Unlike RMT, the HT-Transformer processes a sequence in a single pass while retaining activations from all layers, rather than only the last, for subsequent token processing. In contrast to Longformer, HT-Transformer employs special tokens instead of solely modifying the attention matrix. Moreover, it restricts attention to future tokens, forcing history tokens to accumulate local information.

## 5 EXPERIMENTS

We conduct experiments on datasets spanning multiple domains. The Churn[1], AgePred[2], and Alfabattle[3] datasets represent a range of downstream tasks in the financial domain. The Taobao dataset[4]

---

[1]https://boosters.pro/championship/rosbank1/
[2]https://ods.ai/competitions/sberbank-sirius-lesson
[3]https://boosters.pro/championship/alfabattle2/overview
[4]https://tianchi.aliyun.com/dataset/46

| Dataset | # Seq. | Events | Fields | Mean length | Mean duration | Time unit | Downstream | | |
|---------|--------|--------|--------|-------------|---------------|-----------|------------|--------|--------|
| | | | | | | | Target | Classes | Metric |
| Churn | 10217 | 1M | 6 | 99.3 | 80.5 | Day | Churn | 2 | ROC AUC |
| AgePred | 50000 | 44M | 3 | 875 | 718 | Day | Age group | 4 | Accuracy |
| Alfabattle | 1466527 | 343M | 15 | 234 | 275 | Day | Default | 2 | ROC AUC |
| MIMIC-III | 52103 | 23M | 3 | 407 | 108 | Day | Mortality | 2 | ROC AUC |
| Taobao | 9904 | 5M | 3 | 527 | 12.9 | Day | Activity | 2 | ROC AUC |

Table 1: Datasets statistics

represents user interactions in e-commerce, and MIMIC-III (Johnson et al., 2016) is a widely used collection of medical records. Summary statistics for these datasets are provided in Table 1.

We evaluate three primary baseline approaches: supervised learning, NTP pretraining (Radford et al.), and contrastive learning using CoLES (Babaev et al., 2022). Each method is applied to two backbone architectures. For RNN-based models, we use a GRU backbone (Cho et al., 2014), while Transformer-based models employ a decoder-only architecture (Radford et al.).

All models are trained using the Adam optimizer (DP & J, 2015) with a fixed learning rate of 0.001. The maximum number of training epochs varies by dataset and ranges from 60 to 120. Early stopping is applied based on validation performance to prevent overfitting. Experiments were conducted on NVIDIA A100 GPUs. For all datasets except Alfabattle, training was performed on a single GPU. Due to the larger size of the Alfabattle dataset, some experiments were accelerated using 2 GPUs to reduce training time.

Hyperparameters, including the loss weights for the NTP objective and model size, are optimized using a Bayesian optimizer (Snoek et al., 2012) applied to the NTP RNN baseline. The resulting configurations are reused across all other RNN settings. For Transformer models, we separately tune the number of layers and the hidden dimension using the NTP configuration and apply these settings consistently across all Transformer-based variants.

For each method, we report the mean and standard deviation of evaluation metrics across five different random seeds. An exception is made for the Alfabattle dataset, where three seeds were used due to computational constraints.

To assess the quality of extracted embeddings, we train a gradient boosting classifier for each downstream task using the LightGBM library (Ke et al., 2017). The classifier is trained on frozen embeddings and uses the same hyperparameters as in the CoLES baseline (Babaev et al., 2022).

**Baselines implementation details.** Two of our baselines, Recurrent Memory Transformer Synerise (2025) and Longformer Beltagy et al. (2020), were originally proposed to address the limited scalability of standard Transformers, whose complexity grows quadratically with input length. These models were not specifically designed for embedding extraction, and therefore require additional modifications when applied to classification tasks.

The Recurrent Memory Transformer was originally applied to fixed-length chunks of the input sequence. We follow the same procedure during pretraining but append memory tokens to the end of each chunk for embedding extraction. The final embedding is obtained by averaging activations of the memory tokens.

The Longformer architecture was introduced for bidirectional Transformers, which are incompatible with next-token prediction pretraining. To adapt it, we modify the attention mask by enforcing causality for regular tokens, while allowing global tokens to attend to all tokens, including future ones. Conversely, all tokens can attend to global tokens regardless of relative position. An illustration of our modified Longformer mask is provided in Figure 3c. In addition, we observed that convolutional attention leads to suboptimal performance. Instead, we integrate global tokens directly into the causal attention mask. We further found that using a single global token at the end of each sequence consistently outperforms configurations with multiple tokens placed at regular or random positions. As a result, our final Longformer variant employs a single global token without convolutional masking, and the sequence embedding is obtained by averaging the activations of the last token.

To the best of our knowledge, we at the first time apply Recurrent Memory Transformers and Longformer for embedding extraction in event sequences classification tasks.

## 5.1 CLASSIFICATION OF EVENT SEQUENCES

Classification results are presented in Table 2. Among baselines in the unsupervised setting, the standard NTP Transformer significantly outperforms its RNN counterpart only on the MIMIC-III dataset, while performing worse on Churn, AgePred, and Alfabattle. This highlights the limitations of traditional Transformer architectures in learning compact and informative sequence representations for downstream tasks.

The proposed HT-Transformer effectively addresses these limitations. It consistently outperforms the NTP Transformer in all comparisons. At the same time, contrastively pretrained RNN (CoLES) surpasses HT-Transformer on AgePred and Taobao datasets, however on Taobao the difference is not statistically significant.

The AgePred task differs from the others because it requires predicting a global property, specifically the client's age group, using historical event data. As discussed in the following section, history tokens are designed to capture recent and predictive information, which is beneficial for forecasting future events but less effective for tasks that require encoding global sequence properties. As a result, embeddings extracted from HT-Transformer underperform CoLES RNN.

| Method | Churn | AgePred | Alfabattle | MIMIC-III | Taobao |
|---|---|---|---|---|---|
| Supervised RNN | $79.10 \pm 0.80$ | $61.18 \pm 0.49$ | $76.47 \pm 1.13$ | $91.46 \pm 0.10$ | $84.91 \pm 1.17$ |
| Supervised Transformer | $80.92 \pm 0.66$ | $54.88 \pm 2.37$ | $74.90 \pm 0.08$ | $77.48 \pm 1.22$ | $79.71 \pm 1.68$ |
| NTP RNN | $81.56 \pm 0.59$ | $60.05 \pm 0.29$ | $79.83 \pm 0.05$ | $90.68 \pm 0.07$ | $83.28 \pm 1.42$ |
| NTP Transformer | $80.92 \pm 0.66$ | $56.16 \pm 0.51$ | $78.63 \pm 0.12$ | $91.28 \pm 0.10$ | $83.39 \pm 1.43$ |
| NTP Rec. Mem. Transf. | $80.23 \pm 0.21$ | $58.43 \pm 0.39$ | $80.25 \pm 0.05$ | $91.82 \pm 0.04$ | $80.54 \pm 0.76$ |
| NTP Longformer | $81.48 \pm 0.66$ | $57.64 \pm 0.29$ | $65.91 \pm 0.34$ | $89.26 \pm 0.18$ | $84.76 \pm 1.67$ |
| CoLES RNN | $82.82 \pm 0.28$ | $\mathbf{62.42 \pm 0.33}$ | $79.30 \pm 0.08$ | $87.44 \pm 0.20$ | $\mathbf{85.56 \pm 1.14}$ |
| CoLES Transformer | $78.92 \pm 0.49$ | $59.92 \pm 0.30$ | $78.40 \pm 0.00$ | $87.06 \pm 0.38$ | $82.03 \pm 0.98$ |
| HT-Transformer | $\mathbf{83.34 \pm 0.42}$ | $60.10 \pm 0.39$ | $\mathbf{80.42 \pm 0.12}$ | $\mathbf{92.00 \pm 0.09}$ | $84.65 \pm 1.07$ |
| *Impr. over NTP Transf.* | +2.42 | +3.94 | +1.79 | +0.72 | +1.26 |

Table 2: Pretrained models classification results.

## 5.2 GLOBAL CLASSIFICATION AND FUTURE-ORIENTED TASKS

While the concept of history tokens is broadly applicable, we observe certain limitations when combined with next token prediction during pretraining. The next token prediction objective encourages the model to focus on extracting recent information that is directly relevant for forecasting upcoming events. In contrast, downstream tasks involving classification based on global or persistent properties, such as long-term user characteristics, may benefit more from contrastive pretraining or simpler aggregation strategies, such as averaging Transformer outputs across the sequence.

To investigate this effect, we conduct experiments on a synthetic dataset specifically designed to evaluate the suitability of different representation learning methods for local versus global tasks. In this dataset, we sample ten distinct transition matrices, each defining a Markov process by specifying the probability of transitioning from one label to another. We then construct nonstationary sequences by concatenating multiple segments, each generated using a different transition matrix, as illustrated in Figure 4.

We introduce two classification tasks for our synthetic dataset. The global classification task requires predicting the total number of transition matrices used in a sequence ranging from 1 to 5. This task demands that the model capture information across the entire sequence. In contrast, the local classification task involves identifying the index of the transition matrix used in the final segment, which depends only on the most recent data.

The results of classification experiments with Transformer-based models are reported in Table 3. The NTP Last and NTP Avg baselines correspond to Transformer models with different embedding extraction strategies. NTP Last uses the activations of the final token, whereas NTP Avg computes the average activations across all input tokens. The results show that Last aggregation and

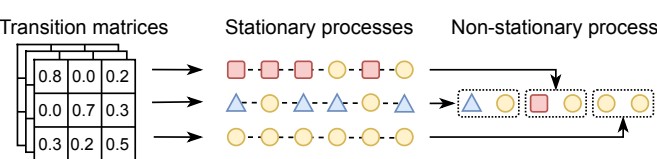

Figure 4: Markovian generative process for the toy dataset.

| Method | Local (Last part) | Global (Num. parts) |
|---|---|---|
| Supervised | 0.71 | 1.00 |
| NTP Last | 0.53 | 0.73 |
| NTP Avg | 0.40 | 0.88 |
| CoLES | 0.33 | **0.94** |
| NTP HT | **0.55** | 0.85 |

Table 3: Toy dataset classification accuracy.

HT-Transformer achieve superior accuracy on the local task, while average aggregation and contrastive pretraining yield the highest accuracy on global classification. These findings suggest that history tokens are particularly effective for tasks that depend on recent context, such as future event prediction, whereas contrastive pretraining and embedding averaging are more suitable for global classification tasks that require holistic sequence understanding

## 5.3 ABLATION STUDIES

**Training Strategies.** In the introduction of HT-Transformer, we outlined alternative strategies for history token placement and selection. Table 4 compares these alternatives with the default HT-Transformer configuration, which employs the Bias-End placement strategy and Last selection of history tokens. The results indicate that the default configuration yields superior downstream performance on considered datasets.

| Method | Churn | MIMIC | Taobao | AVG |
|---|---|---|---|---|
| Uniform pl. | 83.23 | 91.92 | 83.72 | 86.29 |
| Last sel. | 82.92 | 91.90 | 83.78 | 86.20 |
| Bias-End + Random | **83.34** | **92.00** | **84.65** | **86.66** |

Table 4: Comparison of history token placement and selection strategies.

**Hyperparameters.** HT-Transformer introduces two key hyperparameters: the insertion frequency $f$ of history tokens and the application probability $p$. The insertion frequency determines the number of history tokens relative to the input length, while the application probability specifies the proportion of training batches in which history tokens are applied.

Figure 5a shows that the model's performance is not highly sensitive to the exact value of $f$. On the Churn and MIMIC-III datasets, even a single history token achieves performance comparable to that of more frequent insertions. For AgePred and Alfabattle, however, increasing the number of history tokens leads to consistent improvements in performance.

Figure 5b illustrates the effect of varying the application probability $p$. The results indicate that setting $p$ too low significantly degrades performance, with an exception on the Churn dataset. Using the maximum value $p = 1$ on the Churn dataset also results in a modest performance drop. Interestingly, training with $p = 0$ still outperforms a standard NTP Transformer. Our analysis revealed that embeddings of a randomly initialized [CLS] token at the end of the sequence performs better than using the final token's output representation.

Based on these observations, we recommend setting the history token frequency to 10% of the input length and the application probability $p$ to 50%, as used in our default configuration. This setting provides the most stable and consistent performance across all evaluated datasets.

## 6 LIMITATIONS AND FUTURE WORK

In this paper, we demonstrate the effectiveness of using history tokens for event sequence classification. We introduce a new Transformer-based architecture, evaluate multiple design choices, and identify configurations that lead to strong downstream performance across a range of domains. However, several aspects of the method remain open for further exploration.

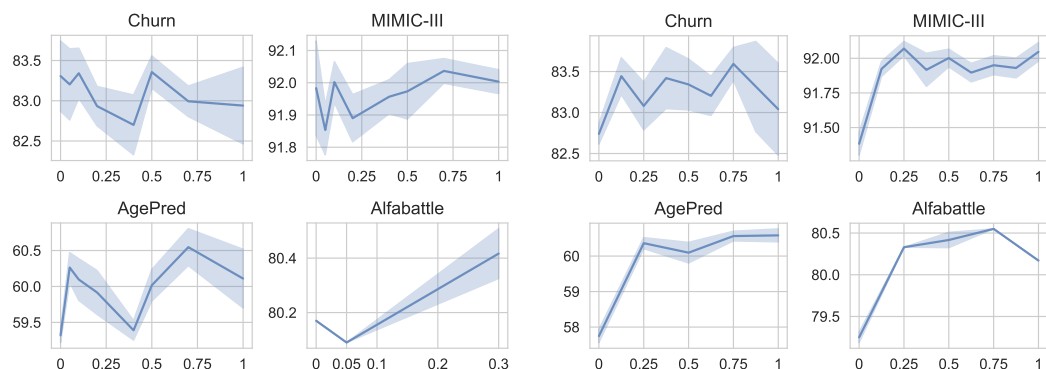

(a) The dependency of the pretraining quality on the history token insertion frequency $f$. Zero frequency corresponds to a single token per sequence.

(b) The dependency of the pretraining quality on the history token application probability $p$.

Figure 5: Evaluation results under different hyperparameter settings.

First, as shown in our ablation studies, the placement of history tokens affects downstream performance. We compared two strategies: uniform placement and the Bias-End approach. Future work could pursue a more detailed analysis of token positioning and sampling policies.

Second, our current implementation relies on the standard PyTorch multi-head attention module. This component may not be optimal for working with the custom attention masks required by the HT-Transformer. Future technical improvements could focus on optimizing the attention mechanism, particularly by exploiting the mask's sparsity. Since only a small subset of tokens participates in the complete self-attention computation, the total computational cost can be reduced. As a result, the HT-Transformer has the potential to offer faster training compared to conventional causal Transformers.

Our experiments demonstrate that HT-Transformer may be suboptimal for global classification tasks. Future research could explore the design of architectures that effectively address both global and local tasks within a unified framework. One promising direction is to combine embeddings extracted from the model using different algorithms, thereby leveraging their complementary strengths.

Overall, the proposed architecture offers a promising direction for highly efficient and accurate modeling of event sequences. Future work can continue to improve the method's predictive performance and computational scalability.

## 7 CONCLUSION

This paper introduced HT-Transformer, a novel architecture designed to enhance Transformer-based models for event sequence classification by explicitly accumulating historical information through learnable history tokens. We identified the inherent limitations of standard Transformers in tasks requiring future event prediction, specifically the lack of a unified representation that effectively captures sequential context. To address this limitation, we proposed a simple yet effective mechanism where history tokens act as information bottlenecks during NTP pretraining, analogous to hidden states in recurrent neural networks.

Our method eliminates the need for auxiliary objectives such as contrastive learning, instead leveraging sparse attention patterns to ensure efficient information aggregation. Extensive empirical evaluations across real-world financial, healthcare, and e-commerce datasets demonstrate that HT-Transformer consistently outperforms conventional Transformer baselines. Moreover, the proposed method achieves the highest accuracy on three datasets, surpassing all competing methods.

Overall, HT-Transformer represents a significant step toward bridging the performance gap between recurrent and Transformer-based models for future-oriented sequence modeling, combining high-quality embeddings with the long-context modeling capabilities of Transformer architectures.

## REPRODUCIBILITY STATEMENT

The full source code, together with training configurations, hyperparameters, and evaluation outputs, is released on GitHub (see reference link in the Abstract) to facilitate reproducibility of our results. All experiments are conducted on publicly available datasets, except MIMIC-III. Access to MIMIC-III requires registration and successful completion of the official ethics training, as mandated by the dataset providers. No proprietary or restricted-access resources were used in this work.

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
