# OpenReview forum: "HT-Transformer: Event Sequences Classification by Accumulating Prefix Information with History Tokens"
_ICLR.cc/2026/Conference — ICLR 2026 Conference Withdrawn Submission_

### Official Review · Reviewer_iarj · 2025-10-21

**Soundness:** 2
**Presentation:** 2
**Contribution:** 2
**Rating:** 2
**Confidence:** 4

**Summary:**

disclaimer : i did not use any LLM for this review.

The paper works on event sequences classification. The authors introduce history token(s) in the pretraining of event sequences by strategically inserting history tokens to a transformer structure for next event prediction task.  Then in finetune (classification of event sequences), they use the history token again as global feature for this downstream task. The authors demonstrate the good empirical results on 5 real datasets compared to a variety of RNN and Transformer variants.

**Strengths:**

Originality: the authors observe the effect of history embedding/token and its success in NLU from Recurrent memory transformer paper, and apply to event sequences, (irregular) time series.  This is a new adaptation.

quality, the authors conduct lot of experiments and variants of HT-transformer on 5 datasets and that support their 4 main claims.

clarity, overall decent – I can follow the structure of the paper fairly well.

significance— I think the history token or in general recurrent memory transformer can be interesting to researcher in time series/event sequences.

**Weaknesses:**

Originality : it can be improved by thinking what are unique aspects of time series/event sequences from the adaptation perspective. It can also be improved by provide some theoretical aspect to explain why bring in history token helps learn global characteristics to predict well on classification

Quality:  it can be improved by extending experiments to braoder setting: including time series, and adding regression etc. Currently it is a little bit weak as the authors did mention time series/ event seuqnces, classification/regression.

Clarity: I am a little confused about PRELIMINARIES ON EVENT SEQUENCES. The authors should separate their contributions and related background information. For example, I am not sure about whether they use eqn (3) as their training objective or not. I know eqn 2 is used in pretraining.

Significance: I think the current scope is a little bit narrow.  Classification on event sequence is less study – there are a few studies on cluster event sequences ( A Dirichlet Mixture Model of Hawkes Processes for Event Sequence Clustering, (neurips 17 ) . researcher in this domain tends to focus on (long horizon/ next event)future prediction/generation, structure learning. Not so much in classification. So the proposed problems are less motivated and the datasets are not standard benchmark (esp. with label). I think the approach is better suited for time series classification/regression.

**Questions:**

1.	Do the authors use eqn 3 in training or pretraining?

2. What does it mean by “ The Uniform strategy inserts history tokens at uniformly sampled positions. However, this approach can lead to a discrepancy between training and inference, as history tokens are typically positioned near the end of the sequence during evaluation “ ?  Inference mean the fine tuning step right? What is the discrepancy between training and inference?

3.The ablation experience is based on markovian generation of type only events. So this is no time step and numerical values correct? So how is training and finetuning different from event sequences with timestamps ?

4. The current version of history token is a little bit strange to me.  One way, it is kind like a mask token like in the pretraining of BERT. in another way, it also kind of act like RNN however, does not change its state. In HT last history token acts like such state so that future token only depend on this state and token after this. I am not sure why it works.  It would be great if the authors can provide some explaination.

---

### Official Review · Reviewer_xrtb · 2025-10-22

**Soundness:** 3
**Presentation:** 3
**Contribution:** 3
**Rating:** 4
**Confidence:** 4

**Summary:**

This paper presents a novel approach to improving temporal sequence prediction by introducing history tokens, dedicated tokens that accumulate and represent historical context. These tokens function as dynamic memory units, allowing the model to construct richer and more informative representations of past events during both pretraining and inference. Furthermore, the authors focus their investigation on two primary design dimensions:

1. Placement: They evaluate two strategies for positioning history tokens: uniform distribution across the sequence and end-biased placement closer to the sequence's conclusion.

2. Attention: They examine how sequence tokens should attend to history tokens, comparing last-token attention with random attention mechanisms.

Extensive experiments on five datasets spanning finance, e-commerce, and healthcare demonstrate that the proposed HT-Transformer consistently outperforms strong baselines, including RNNs, standard Transformers, Longformer, and Recurrent Memory Transformer (RMT) in different settings. These results underscore the value of integrating history tokens into transformer-based architectures for temporal and event sequence modeling.

**Strengths:**

1. Novelty and Practicality: This paper introduces a simple yet effective idea called history tokens to flexibly incorporate historical context addressing a key limitation in temporal sequence modeling.  The concept of history token is very interesting.

2. Well Explored Design Choices: Authors  provide a systematic study of token placement and attention strategies with solid ablation experiments that enhance both interpretability and adaptability.

3. Strong Empirical Results : Experiments demonstrate consistent performance gains over RNNs Transformers Longformer and RMT across five datasets in finance healthcare and e-commerce and so on.

4. Reproducibility : Public code are available supporting transparency and facilitating future research.

**Weaknesses:**

While this paper presents valuable contributions, addressing the following points could further strengthen its impact and clarity:

1. Clarify the Relationship with CLS Tokens:  It would be more comprehensive to include a detailed comparison between history tokens and the CLS token mechanism used in models like BERT, other than the history tokens appearing multiple times in a sequence. Elaborating on how they differ functionally and architecturally will help clarify history token's unique role.

2. Compare with Compressed Vector Methods in Long-Context Transformers:  Providing a methodological comparison between history tokens and compressed or pooled vector approaches used in long-context models such as Unlimiformer, Linformer, Reformer, sparse/blockwise attention mechanisms, and so on, would contextualize the contribution and highlight advantages of history tokens.

3. Broaden Experimental Baselines:  Incorporating additional baselines like BigBird, FlashAttention, ModernBERT, Mamba, and Structured State Space Models (SSMs) would strengthen the empirical evaluation and better position the work within the current state of the art.

4. Include Efficiency and Computational Cost Analysis:  Adding an analysis of computational efficiency and resource usage compared to other long-sequence models is important for assessing the practical applicability of the approach. Including such details in the main text will provide readers with a clearer understanding of its scalability.

Addressing these points would provide greater clarity, strengthen the motivation, and improve the overall impact of the work.

**Questions:**

1. How does the HT-Transformer relate to traditional time series forecasting methods and recent models like PatchTST? Could it be applied to classic forecasting tasks such as next-step prediction or multivariate forecasting, and how well does it capture inductive patterns like seasonality, trends, or locality?

2. What factors influence the optimal number or ratio of history tokens relative to the input length? Are there cases where using too many tokens might introduce redundancy or reduce attention efficiency?

3. Could the placement of history tokens be adapted to the structure of the data or task, for example, aligning with seasonal events, or paragraph or document boundaries in text? Would such task-aware placement improve efficiency or performance?

---

### Official Review · Reviewer_xmm2 · 2025-10-28

**Soundness:** 2
**Presentation:** 2
**Contribution:** 1
**Rating:** 2
**Confidence:** 4

**Summary:**

This paper proposes HT-Transformer, a transformer-based architecture designed for irregul sequence classification. The key idea is to introduce history tokens to accumulate prefix (historical) information during pretraining. The accumulated history token serves as a meaningful representation for downstream tasks. Moreover, it proposes a contrastive pretraining method for next token prediction and cont. The authors evaluate HT-Transformer on diverse datasets spanning finance, e-commerce, and healthcare domains, indicating strong performance compared with autoregressive Transfomers with next token prediction task.

**Strengths:**

1. It proposes “history tokens” as accumulators of prefix informationthat mimics the RNN hidden-state mechanism.

2. Strong empirical evidence: The method is evaluated across multiple domains.

**Weaknesses:**

1. Lack of methodological contribution: This work proposes "history token" which is very similar to [SEP] and [EOS] token in BERT, which can be used to summarize the past information.

2. Token aggregation: For classification, this work does not compare or make it clear why history token is better than traditional [EOS] or simply avg pooling. It seems like, if you use history token at the end of sequence (Fig 1.b), you actually use the [EOS] token which is already used by GPT before.

3. Method Comparsion: it does not compare with recurrent (linear) transformers, which can be considered as deep RNNs.

**Questions:**

1. How does the proposed “history token” mechanism differ functionally from the state memory variable? In linear transformer model, it will compute the state variable using K and V matrix, which has dxd size. This state variable is larger history token you propose, and it is dynamic. So each token's state variable can summarize all the past information. It seems much flexiable and representative than introducing an extra specical token.

2. This work actually has 2 loss L_MAE and L_CONT, it is not clear how to combine them?

3. Did you evaluate how many history tokens are actually attended to during pretraining? This would help understand if the model genuinely learns to use them.

---

### Official Review · Reviewer_65ji · 2025-10-30

**Soundness:** 2
**Presentation:** 3
**Contribution:** 2
**Rating:** 2
**Confidence:** 3

**Summary:**

This paper introduces HT-Transformer, a variant of the transformer architecture designed for event sequence classification. The core contribution is the use of history tokens, which learns to extract cumulative prefix information during pretraining. The authors show how this approach enables transformers to better represent sequential context for downstream tasks, especially for predicting future events. Extensive empirical evaluations are reported across datasets in different domains, with comparisons against multiple methods, including RNNs, NTP transformers, and adapted techniques such as the recurrent memory transformer and Longformer.

**Strengths:**

**Clear motivation and conceptual description:**
The paper clearly explains the limitations of the existing transformers and presents the differences among the proposed HT-Transformer and previous variants, especially on the masking strategy. Detailed descriptions of the history token insertion strategies are provided.

**Comprehensive experiment:**
The paper evaluates across multiple domains and covers different methods (contrastive learning, NTP, etc.) on both network architectures (RNN, Transformer).

**Study on Limitations:**
Experimental results show that the proposed HT-Transformer performs worse on the global task.

**Weaknesses:**

1. The pretraining part should have more details. Currently, Sec. 2 includes part of the training objectives and Sec. 3.1 introduces the masking and inserting strategies. However, it is still not clear what the optimization target is for this pretraining stage. This is important as the pretrained embeddings are directly used for classification.
2. The performance gain is marginal, considering that LongFormer also aims to reduce memory. The implementation details and ablation study actually weaken the contribution: a step increase is observed in the low probability zone, which is counterintuitive if the history token actually has a significant effect on the performance.

**Questions:**

Weakness 1:  How are the inserted history tokens supervised during the "next event prediction" modelling framework?
Weakness 1&2: In the insertion strategy part, when a sample is selected as "no application", do you mean there are no history tokens in the middle of the sequence (excluding the final one), or there are no history tokens throughout the whole sequence?
Weakness 2: Could you further explain the phenomenon of the step increase in $p=0$ to $p=0.25$?

---

### Official Review · Reviewer_fKHk · 2025-10-31

**Soundness:** 2
**Presentation:** 2
**Contribution:** 2
**Rating:** 2
**Confidence:** 3

**Summary:**

The paper proposes HT-Transformer, a novel Transformer architecture for event sequence classification that introduces learnable history tokens to accumulate contextual information through sparse attention patterns during next-token prediction pretraining. The method eliminates the need for auxiliary objectives like contrastive learning and demonstrates strong performance across financial, healthcare, and e-commerce datasets.

**Strengths:**

(1). The concept of history tokens with specialized attention patterns thoughtfully integrates recurrent principles into Transformer architectures.

(2). The writing is generally clear, with the method explained step-by-step.

**Weaknesses:**

(1). The paper positions HT-Transformer as a significant departure from prior work, but the core idea of using special tokens to aggregate sequence information has been explored in existing works. The authors should more clearly differentiate their contribution.

(2). The paper lacks explanation for why the Random strategy and Bias-End placement work better. It's better to provide analysis about why these strategies improve performance.

(3). The paper uses gradient boosting on frozen embeddings for classification. It is unclear why this choice was made and whether it is standard.

**Questions:**

(1). Why does the Random history token selection strategy consistently outperform the Last strategy?

(2). The paper mentions that the Longformer was adapted for causal attention, but the details are sparse. How exactly was the attention mask modified?

(3). The paper uses the average of hidden activations of the history token as the sequence embedding. Was any other aggregation method (e.g., max pooling, concatenation) considered and why?

(4). Are there particular event sequence characteristics (temporal density, categorical diversity) where HT-Transformer shows weak performance?

**Details Of Ethics Concerns:**

-

---

### Note · Authors · 2025-12-02

I have read and agree with the venue's withdrawal policy on behalf of myself and my co-authors.